# The purine nucleoside phosphorylase *pnp-1* regulates epithelial cell resistance to infection in *C. elegans*

Eillen Tecle[1], Crystal B. Chhan[1], Latisha Franklin[2], Ryan S. Underwood[1], Wendy Hanna-Rose[2], Emily R. Troemel[1] *

1 Division of Biological Sciences, University of California, San Diego, La Jolla, California, United States of America, 2 Department of Biochemistry and Molecular Biology, The Pennsylvania State University, University Park, Pennsylvania, United States of America

* etroemel@ucsd.edu

**Data Availability Statement:** RNA-seq read files are available from the NCBI GEO database (accession number GSE165786).

## Abstract

Intestinal epithelial cells are subject to attack by a diverse array of microbes, including intracellular as well as extracellular pathogens. While defense in epithelial cells can be triggered by pattern recognition receptor-mediated detection of microbe-associated molecular patterns, there is much to be learned about how they sense infection via perturbations of host physiology, which often occur during infection. A recently described host defense response in the nematode *C. elegans* called the Intracellular Pathogen Response (IPR) can be triggered by infection with diverse natural intracellular pathogens, as well as by perturbations to protein homeostasis. From a forward genetic screen, we identified the *C. elegans* ortholog of purine nucleoside phosphorylase *pnp-1* as a negative regulator of IPR gene expression, as well as a negative regulator of genes induced by extracellular pathogens. Accordingly, *pnp-1* mutants have resistance to both intracellular and extracellular pathogens. Metabolomics analysis indicates that *C. elegans pnp-1* likely has enzymatic activity similar to its human ortholog, serving to convert purine nucleosides into free bases. Classic genetic studies have shown how mutations in human purine nucleoside phosphorylase cause immunodeficiency due to T-cell dysfunction. Here we show that *C. elegans pnp-1* acts in intestinal epithelial cells to regulate defense. Altogether, these results indicate that perturbations in purine metabolism are likely monitored as a cue to promote defense against epithelial infection in the nematode *C. elegans*.

## Author summary

All life requires purine nucleotides. However, obligate intracellular pathogens are incapable of generating their own purine nucleotides and thus have evolved strategies to steal these nucleotides from host cells in order to support their growth and replication. Using the small roundworm *C. elegans*, we show that infection with natural obligate intracellular pathogens is impaired by loss of *pnp-1*, the *C. elegans* ortholog of the vertebrate purine nucleoside phosphorylase (PNP), which is an enzyme involved in salvaging purines. Loss

**Funding:** This work was supported by National Institutes of Health (www.nih.gov) under R01 AG052622 and GM114139 to ERT, and by National Institute of General Medical Sciences (www.nigms.nih.gov) / National Institutes of Health (www.nih.gov) award K12GM068524 to ET. This publication includes data generated at the UC San Diego IGM Genomics Center utilizing an Illumina NovaSeq 6000 that was purchased with funding from a National Institutes of Health SIG grant (#S10 OD026929). The funders had no role in study design, data collection and analysis, decision to publish, or preparation of the manuscript.

**Competing interests:** The authors have declared that no competing interests exist.

of *pnp-1* leads to altered levels of purine nucleotide precursors and increased expression of Intracellular Pathogen Response genes, which are induced by viral and fungal intracellular pathogens of *C. elegans*. In addition, we find that loss of *pnp-1* increases resistance to extracellular pathogen infection and increases expression of genes involved in extracellular pathogen defense. Interestingly, studies from 1975 found that mutations in human PNP impair T-cell immunity, whereas our findings here indicate *C. elegans pnp-1* regulates intestinal epithelial immunity. Overall, our work indicates that host purine homeostasis regulates resistance to both intracellular and extracellular pathogen infection.

## Introduction

Obligate intracellular pathogens are completely dependent on their hosts for replication. In most cases, these pathogens lack biosynthetic pathways and thus rely on host cells to obtain building blocks for growth, including nucleotides, amino acids and lipids. In particular, viruses are completely reliant on host nucleotides for replication and transcription. As such, host restriction factors can serve to limit the pool of nucleotides available to viruses and thus block their growth. For example, the human SAM domain and HD domain-containing protein 1 is a restriction factor for Human Immunodeficiency Virus (HIV), acting to degrade the deoxynucleotides needed for reverse transcription of the HIV genome [1]. Even eukaryotic pathogens, such as microsporidia, appear to lack nucleotide biosynthetic pathways and instead 'steal' nucleotides from host cells [2,3]. In particular, microsporidia express cell-surface ATP/GTP transporters, which are believed to import host purine nucleotides to support parasite growth and proliferation [4–6].

Microsporidia comprise a phylum of obligate intracellular pathogens related to fungi, with over 1400 species identified [7]. Microsporidia are extremely prevalent in nature and almost all animals are susceptible to infection by at least one microsporidia species [8,9]. Recent work indicates that these fungal pathogens are the most common cause of infection in the wild for the model nematode *C. elegans* and other related nematodes [10,11]. The microsporidian *Nematocida parisii* is the pathogen species most often found to infect wild *C. elegans*, and this pathogen goes through its entire replicative life cycle inside the intestine [10]. Interestingly, the host transcriptional response to *N. parisii* appears to be almost identical to the host response to the Orsay virus, which is a 3-gene positive-sense RNA virus and another natural intracellular pathogen of the *C. elegans* intestine [10,12–14]. This common transcriptional response has been named the Intracellular Pathogen Response, or IPR, and it appears to constitute a novel defense pathway in *C. elegans* [15]. Although the host sensor for *N. parisii* is not known, we have recently discovered that *drh-1*, a *C. elegans* homolog of the mammalian RIG-I dsRNA sensor, mediates induction of the IPR likely through detecting dsRNA or other RNA replication products of the Orsay virus [16].

Forward genetic studies identified *pals-22* and *pals-25* as antagonistic paralogs that regulate the IPR and associated phenotypes [15,17]. *pals-22* and *pals-25* belong to the *pals* gene family in *C. elegans*, which contains at least 39 *pals* genes named for the loosely conserved ALS2CR12 protein signature located in the single ALS2CR12 gene found in each of the human and mouse genomes [15,18,19]. The biochemical functions of ALS2CR12 and *C. elegans pals* genes are unknown, but *pals-22* and *pals-25* appear to dramatically rewire *C. elegans* physiology. *pals-22* mutants have constitutive expression of IPR genes in the absence of infection, and have improved tolerance of proteotoxic stress, as well as increased resistance against *N. parisii* and the Orsay virus, but decreased resistance against the bacterial extracellular pathogen

*Pseudomonas aeruginosa* [15,17]. A mutation in *pals-25* reverses these phenotypes found in *pals-22* mutants, as *pals-22 pals-25* double mutants have phenotypes similar to wild-type animals, including wild-type levels of IPR gene expression. There are approximately 100 IPR genes, which include other *pals* genes like *pals-5* (although notably not *pals-22* and *pals-25*), as well as components of a cullin RING ubiquitin ligase complex, which is required for the increased tolerance of proteotoxic stress observed in *pals-22* mutants [17,20].

To gain insight into how *C. elegans* regulate IPR gene expression and related phenotypes, we sought to identify additional regulators of the IPR. Here, we report that *C. elegans pnp-1* is a novel repressor of the IPR. *pnp-1* is the *C. elegans* ortholog of the vertebrate purine nucleoside phosphorylase (PNP), which functions in the purine salvage pathway to regulate levels of purine nucleotides. Interestingly, mutations in human PNP lead to severe combined immunodeficiency disease due to T-cell dysfunction [21–23]. Our results indicate that, like vertebrate PNP, *C. elegans pnp-1* functions as a purine nucleoside phosphorylase, regulating levels of purine metabolites. We find that *pnp-1* mutants, similar to *pals-22* mutants, display resistance to various natural intestinal pathogens. Surprisingly, unlike *pals-22*, *pnp-1* also negatively regulates the expression of genes that are induced by bacterial infection and by other immune regulators. Moreover, *pnp-1* mutants display resistance to the extracellular bacterial pathogen *P. aeruginosa*. Epistasis analysis indicates that the p38 MAP kinase *pmk-1* is required for the increased resistance of *pnp-1* mutants against extracellular pathogens. In summary, our work indicates that *pnp-1* is a new regulator of the response to both intracellular and extracellular infection, suggesting that purine metabolite levels are important regulators of the response to pathogen infection.

## Results

### *pnp-1* is a negative regulator of IPR gene expression

To identify negative regulators of the IPR, we performed a forward mutagenesis screen using the established *pals-5p*::*gfp* transcriptional reporter, which induces GFP expression in the intestine upon *N. parisii* or Orsay virus infection. From this screen, we isolated mutants with constitutive *pals-5p*::*gfp* expression including the allele *jy90*. After back-crossing *jy90* mutants and mapping the mutation to Chromosome IV, we performed whole genome sequencing to identify the causative allele (S1 Table). From this analysis, we identified a missense mutation in the PNP gene *pnp-1*, which should result in substitution of a conserved serine (S51 or S68 in isoform a or b, respectively) to leucine (Fig 1A). This serine is conserved across phylogeny and has been shown to be required for enzymatic activity of human PNP (S1 Fig) [24]. To confirm that a mutation in *pnp-1* can induce *pals-5p*::*gfp* expression, we used CRISPR/Cas9 editing to generate a deletion allele of *pnp-1* called *jy121*. We found that *pnp-1(jy121)* mutants have constitutive *pals-5p*::*gfp* expression similar to *pnp-1(jy90)* mutants, confirming that *pnp-1* regulates expression of *pals-5p*::*gfp* (Fig 1B–1D). To test if *pnp-1* regulates expression of endogenous *pals-5* mRNA and not just the *pals-5p*::*gfp* transgene, we performed qRT-PCR and analyzed the mRNA expression of *pals-5* and other IPR genes. Here, we found that *pnp-1* mutants constitutively express endogenous mRNA for *pals-5* and other IPR genes in the absence of any IPR trigger (Fig 1E), indicating that wild-type *pnp-1* negatively regulates the mRNA expression of several IPR genes.

Although *pnp-1* mutants, like *pals-22* mutants, constitutively express IPR genes, we find that the levels of IPR gene expression in *pnp-1* mutants are lower than that of *pals-22* mutants (S2A Fig). In addition, we find that loss of *pals-25* does not suppress IPR gene expression in *pnp-1* mutants (S2B Fig) as it does in *pals-22* mutants [17]. Therefore, our data indicates that

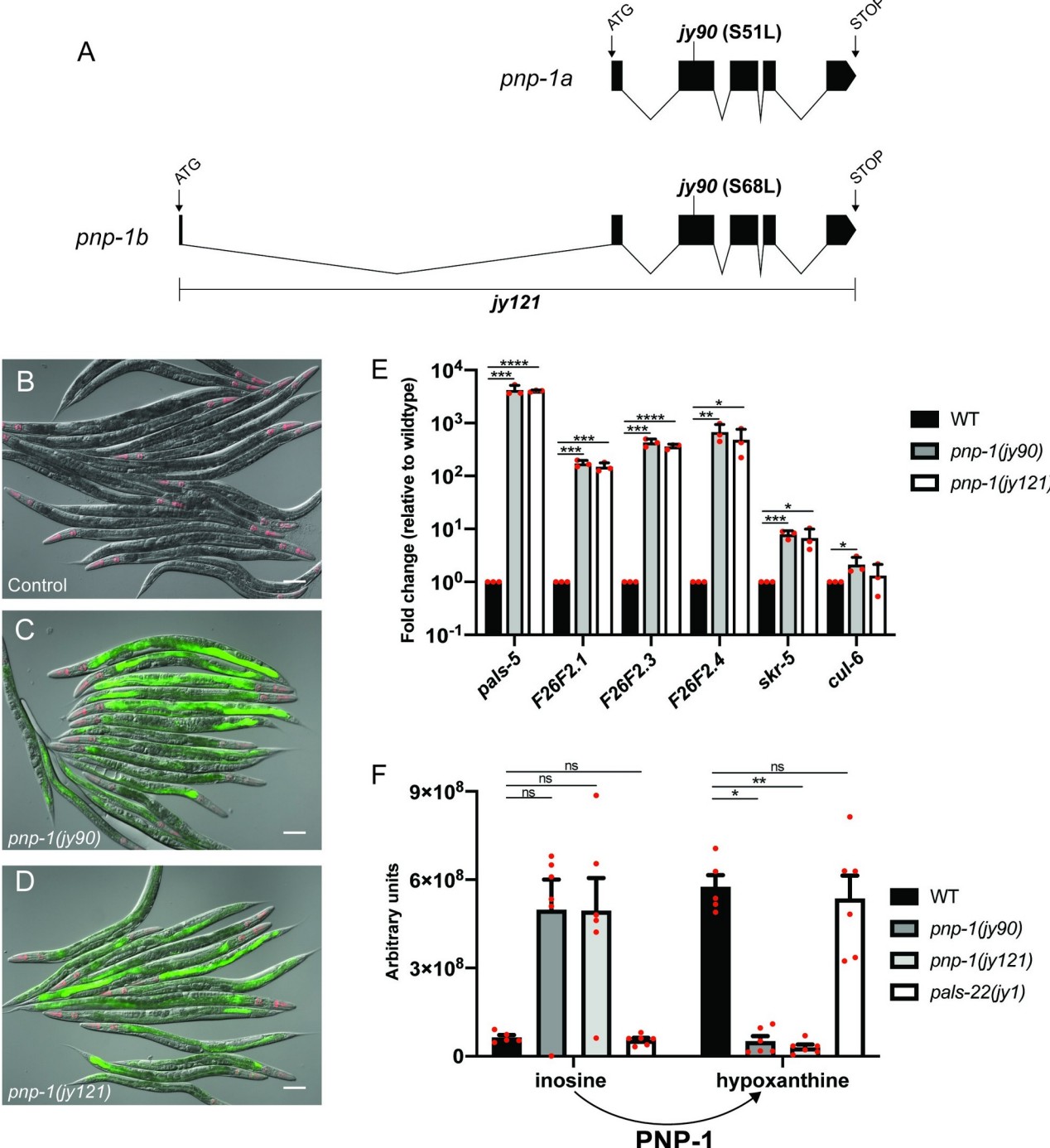

**Fig 1. *pnp-1* mutants have increased expression of IPR genes.** A) Gene structure of the two isoforms of *pnp-1* with exons indicated as black boxes. 5' and 3' untranslated regions are not shown. B-D) *pals-5p::gfp* IPR reporter expression in wild-type animals, *pnp-1(jy90)* and *pnp-1(jy121)* mutants. *myo-2p::mCherry* is a pharyngeal marker for the presence of the IPR reporter transgene. Scale bar is 100 μm. E) qRT-PCR of a subset of IPR genes in *pnp-1 (jy90)* and *pnp-1(jy121)* mutants. Fold change in gene expression is shown relative to wild-type animals. Graph shows the mean fold change of three independent experiments. Error bars are standard deviation (SD). Mixed stage populations of animals were used. **** indicates p < 0.0001 by one-tailed t-test. F) Quantification of inosine and hypoxanthine levels in *pnp-1* and *pals-22* mutants from metabolomics analysis. Graph shows the mean levels of metabolites from six independent experiments for *pnp-1(jy121)*, *pnp-1(jy90)* and *pals-22(jy1)* mutants, and five independent experiments for wild-type animals. Error bars are standard error of the mean (SEM). ** indicates p < 0.01 by the Kruskal-Wallis test. E, F) Red dots indicate values from individual experiments. See materials and methods for more information.

*pnp-1* likely acts in parallel to the more potent *pals-22*/*pals-25* pathway to regulate IPR gene expression.

## *pnp-1* mutants have altered levels of purine metabolites

Vertebrate PNP functions in the purine salvage pathway where purine nucleotides sequentially are degraded to nucleosides and purine bases. These bases are then converted back into purine nucleotides. Specifically, PNP converts the nucleosides inosine or guanosine into the bases hypoxanthine or guanine, respectively (S3A Fig). If *pnp-1* were functioning in *C. elegans* as a PNP, *pnp-1* mutants should have higher levels of nucleosides, and lower levels of purine bases. To determine whether this is the case, we performed targeted Liquid Chromatography-Mass Spectrometry metabolomic analysis to quantify these metabolites in *pnp-1* mutants. Indeed, this analysis revealed that *pnp-1* mutants have higher levels of inosine and lower levels of hypoxanthine as compared to wild-type animals (Fig 1F). However, guanine and guanosine were below detectable levels in both mutants and wild-type animals, so comparisons could not be made for these metabolites. No significant changes were found in the levels of any other detected metabolites of the purine salvage pathway or metabolites of the *de novo* purine synthesis pathway in *pnp-1* mutants (S3 Fig). Of note, the levels of inosine and hypoxanthine in *pals-22* mutants were not significantly different than those of wild-type animals, indicating that not all mutants with constitutive IPR gene expression have altered inosine and hypoxanthine levels. In summary, these findings indicate that, similar to its vertebrate ortholog, *C. elegans pnp-1* functions as a PNP to convert purine nucleosides into free purine bases.

In order to determine whether altered levels of inosine or hypoxanthine may be responsible for altered IPR gene expression, we performed dietary supplementation to investigate whether inosine may cause activation of the *pals-5p*::*gfp* reporter in a wild-type background, or whether hypoxanthine may cause suppression of the *pals-5p*::*gfp* reporter in a *pnp-1* mutant background. Here, we did not see any obvious changes of *pals-5p*::*gfp* expression in either case (see Materials and Methods for more detail). Because we do not have controls to demonstrate uptake of these metabolites into *C. elegans* cells however, it is difficult to make conclusions from these negative results.

## *pnp-1* mutants have increased resistance to intracellular pathogen infection

As *pnp-1* mutants have constitutive expression of IPR genes, we hypothesized that these mutants should be resistant to intracellular pathogen infection. Therefore, we assessed the pathogen load of Orsay virus and *N. parisii* in *pnp-1* mutants. Here we found that Orsay viral load, as determined by qRT-PCR for the viral genome segment RNA1, is significantly lower in *pnp-1* mutants as compared to wild-type animals (Fig 2A). We next analyzed *N. parisii* pathogen load at 3 hours post infection (hpi) by counting individual sporoplasms, which are individual parasite cells likely to represent individual invasion events into intestinal cells [25]. Here we found that *pnp-1* mutants display a significant reduction of the number of sporoplasms per animal as compared to wild-type animals (Fig 2B). We also investigated *N. parisii* pathogen load at 30 hpi, when sporoplasms have developed into replicative meronts within the intestine. At 30 hpi as well, we found that *N. parisii* pathogen load of *pnp-1* mutants is significantly lower than that of wild-type animals (Fig 2C). Of note, we found that *pals-22* mutants have significantly higher resistance to intracellular pathogen infection than *pnp-1* mutants (Fig 2A–2C), perhaps because of the higher levels of IPR gene expression in *pals-22* mutants (S2A Fig).

Consistent with *pnp-1* mutants having lower *N. parisii* pathogen load compared to wild-type animals, we also found that they have increased survival upon infection

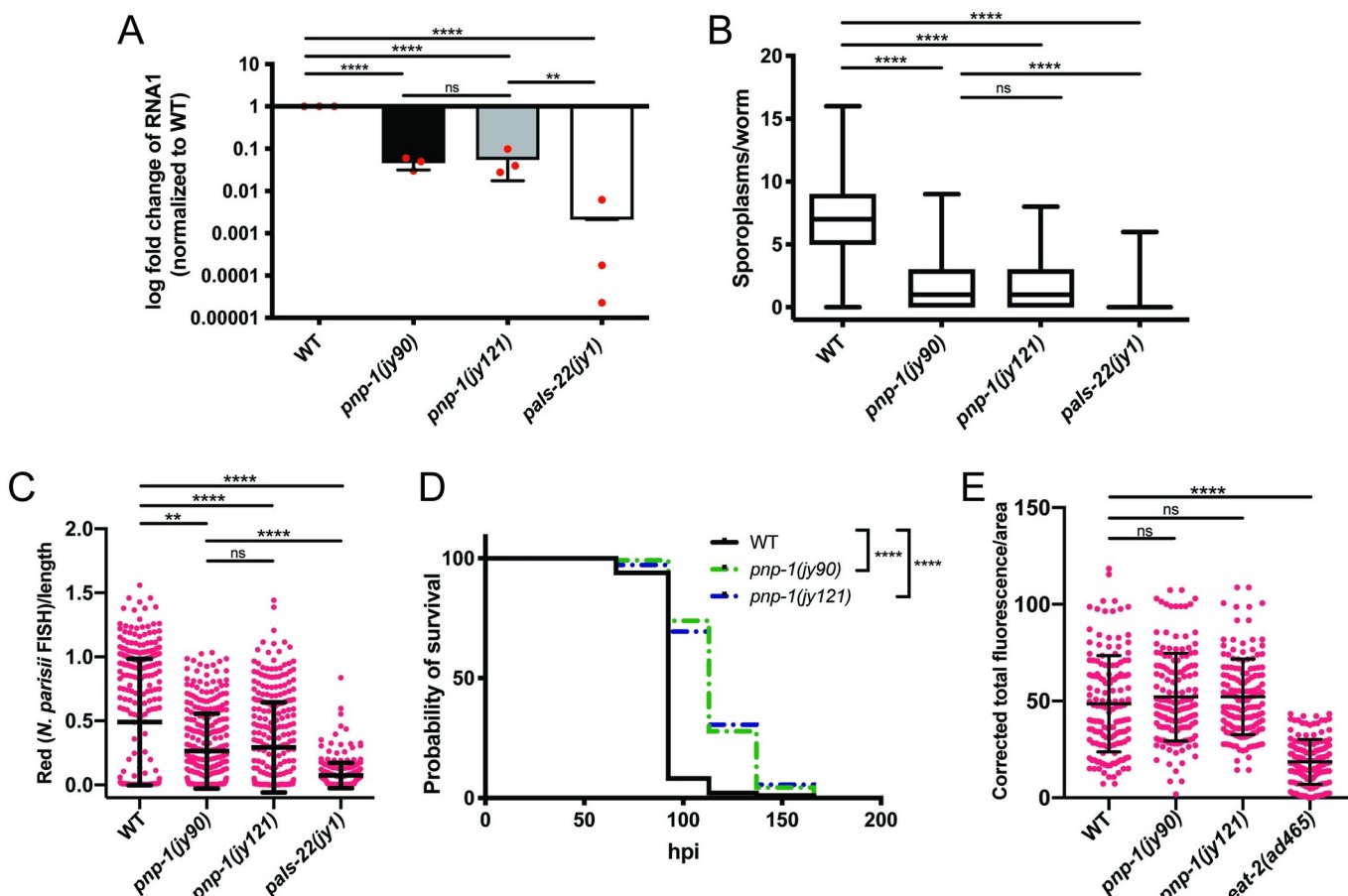

**Fig 2. *pnp-1* regulates intracellular pathogen resistance.** A) qRT-PCR for Orsay viral load in wild-type animals, *pnp-1(jy90)*, *pnp-1(jy121)* and *pals-22(jy1)* mutants. Fold change in gene expression is shown relative to wild-type animals. Graph shows results of three independent experiments. Mean is shown with error bars as SD. Red dots indicate values from individual experiments. Synchronized fourth larval stage (L4) animals were used. **** indicates p < 0.0001 by a one-tailed t-test. B) Quantification of *N. parisii* sporoplasm number in wild-type animals, *pnp-1(jy90)*, *pnp-1(jy121)* and *pals-22(jy1)* first larval stage (L1) mutants at 3 hpi. n = 225 animals per genotype. Box represents 50% of the data closest to the median while whiskers span the values outside the box. C) Quantification of *N. parisii* pathogen load in wild-type animals, *pnp-1(jy90)*, *pnp-1(jy121)* and *pals-22(jy1)* L1 mutants at 30 hpi. n = 300 animals per genotype. *N. parisii* load per animal was quantified with the COPAS Biosort machine and normalized to time-of-flight as proxy for the length of the animal. Each dot represents an individual animal. Mean is shown with error bars as SD. B, C) *N. parisii* was visualized using an *N. parisii* rRNA specific probe. Each graph shows the combined results of three independent experiments. D) Survival of wild-type, *pnp-1(jy90)* and *pnp-1(jy121)* mutants after infection with *N. parisii*. n = 120 per genotype. One experiment of three independent experiments is shown (see S4 Fig for additional two experiments). **** indicates p < 0.0001 by the Log-rank (Mantel-Cox) test. E) Quantification of fluorescent bead accumulation in wild-type animals, *pnp-1(jy90)*, *pnp-1(jy121)* and *eat-2(ad465)* mutants. n = 150 animals per genotype. The corrected total fluorescence per worm was calculated and normalized to worm area. Each dot represents an individual animal. B, C and E) **** indicates p < 0.0001 by the Kruskal-Wallis test. Graphs show combined results of three independent experiments.

compared to wild-type animals (Figs 2D and S4). To address the concern that the pathogen resistance of *pnp-1* mutants is due to feeding defects that lower the exposure of these animals to intestinal pathogens, we fed them fluorescent beads and quantified accumulation in the intestinal lumen. We found that accumulation of these beads is not significantly different in either of the *pnp-1* mutants as compared to wild-type animals, whereas the known feeding-defective *eat-2* mutants displayed significantly less bead accumulation compared to wild-type animals (Fig 2E). Taken together, these results indicate that wild-type *pnp-1* functions to negatively regulate resistance to and survival upon intracellular pathogen infection.

### *pnp-1* functions in the intestine to regulate the IPR

To determine the site of action for *pnp-1*-mediated regulation of the IPR, we investigated its tissue expression using a "TransgeneOme" construct containing PNP-1 tagged at the C terminus with GFP and 3×FLAG, surrounded by a ~20-kb endogenous genomic regulatory region [26]. We generated transgenic animals containing this construct and observed GFP expression in the 20 epithelial cells that comprise the intestine, as well as in several head neurons (Figs 3A

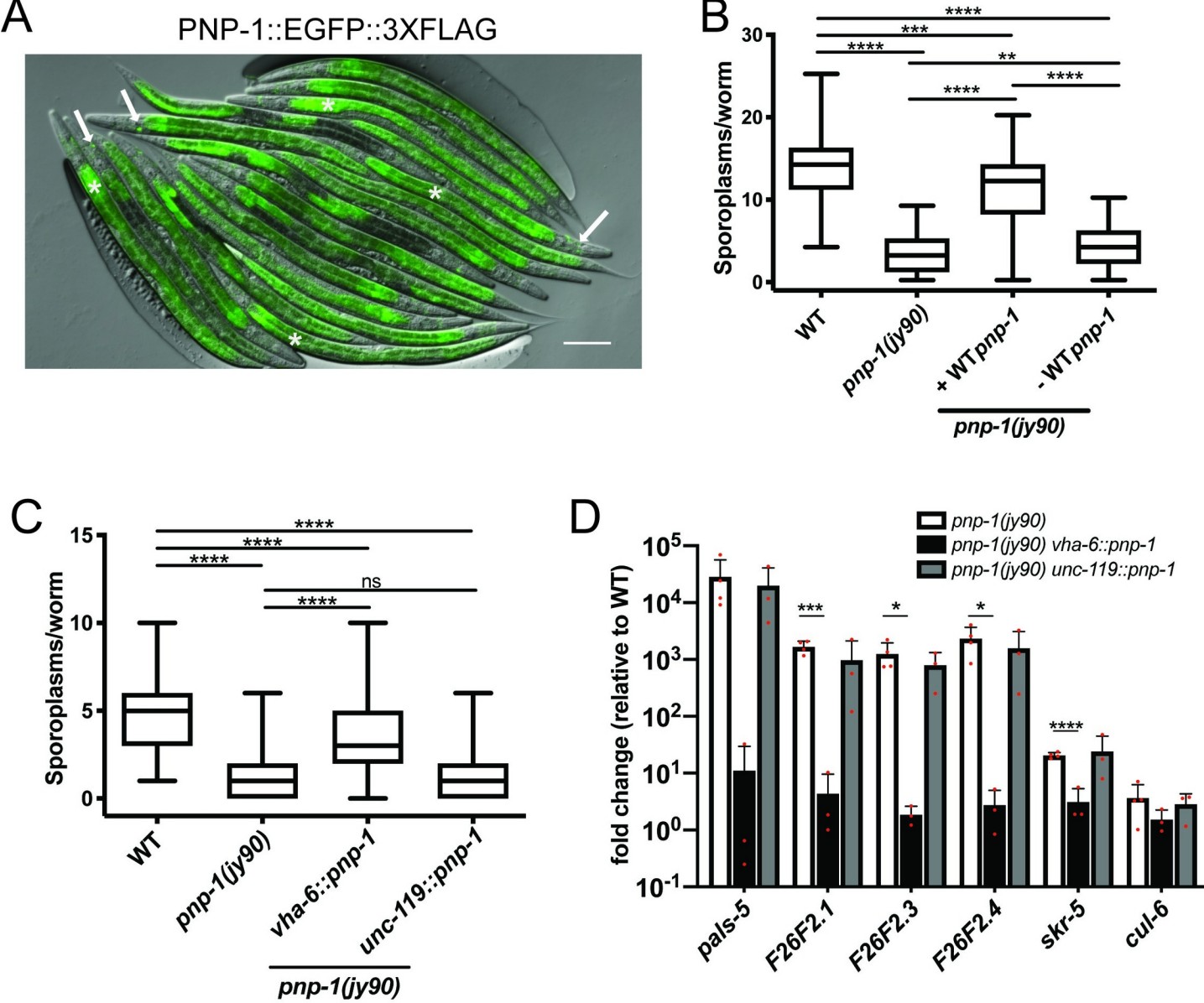

**Fig 3. *pnp-1* functions in the intestine to regulate the IPR.** A) Expression of PNP-1::EGFP::3XFLAG under control of the wild-type *pnp-1* genomic locus. Asterisks indicate intestines and arrows indicate neurons. Scale bar is 100 μm. B) Quantification of *N. parisii* sporoplasm number at 3 hpi in *pnp-1* mutants containing the rescuing *pnp-1::egfp::3Xflag* genomic locus in L1 animals (indicated as "+WT *pnp-1*"), as well as their non-transgenic siblings (indicated as "-WT *pnp-1*"). n = 255 animals per genotype. C) Quantification of *N. parisii* sporoplasm number at 3 hpi in *pnp-1* mutants containing wild-type *pnp-1* cDNA under the control the *vha-6* (intestinal) or *unc-119* (neuronal) promoters in L1 animals. n = 150 per genotype. B, C) *N. parisii* was visualized using an *N. parisii* rRNA specific probe. Each graph shows the combined results of three independent experiments. In the graphs, the box represents the 50% of the data closest to the median while the whiskers span the values outside the box. **** indicates p < 0.0001 by the Kruskal-Wallis test. D) qRT-PCR of a subset of IPR genes in adult *pnp-1* mutants containing wild-type *pnp-1* cDNA under the control the *vha-6* or *unc-119* promoter. Fold change in gene expression is shown relative to control. Graphs show the combined results of three independent experiments. Red dots indicate values from individual experiments. **** indicates p < 0.0001 by a one-tailed t-test.

and S5). Importantly, expression of this PNP-1::GFP transgene rescued the decreased number of sporoplasms in *pnp-1* mutants (Fig 3B), supporting the model that expression from this transgene reflects endogenous PNP-1 expression.

Next, we used single-copy tissue-specific expression to investigate where *pnp-1* regulates IPR phenotypes. In a *pnp-1* mutant background, we generated single copy insertions of *pnp-1* under the control of an intestine-specific promoter (*vha-6*) or a neuron-specific promoter (*unc-119*) into the same genomic locus. Expression of *pnp-1* in the intestine, but not in neurons, rescued the *pnp-1* mutant phenotypes of decreased number of sporoplasms (Fig 3C) and increased expression of IPR genes (Fig 3D). Altogether, these results demonstrate that *pnp-1* acts in intestinal epithelial cells to regulate IPR gene expression and pathogen resistance.

## *pnp-1* mutants display phenotypes not previously associated with IPR activation

As *pnp-1* had not been characterized before in *C. elegans*, we next explored additional phenotypes. We focused on phenotypes found in *pals-22* mutants, as these mutants have constitutive IPR expression, like *pnp-1* mutants [15,17]. First, we determined the lifespan of *pnp-1* mutants. In contrast to the short-lived *pals-22* mutants, we found that the lifespan of *pnp-1* mutants appears similar to wild-type animals (Figs 4A and S6). Second, we investigated the thermotolerance of *pnp-1* mutants, because *pals-22* mutants display increased resistance to proteotoxic stress, including better thermotolerance compared to wild-type animals. Here, *pnp-1* mutants also had a distinct phenotype from *pals-22* mutants, as they displayed significantly decreased thermotolerance as compared to wild-type animals (Fig 4B). Interestingly, *pnp-1* appears to be acting downstream or in parallel of *pals-22* for thermotolerance, as the *pals-22*; *pnp-1* double mutants show decreased thermotolerance, similar to *pnp-1* single mutants (Fig 4C).

We further explored the pathogen resistance phenotypes of *pnp-1* mutants in comparison to *pals-22* mutants. Previous work had shown that *pals-22* mutants have increased susceptibility to the extracellular Gram-negative bacterial pathogen *P. aeruginosa* strain PA14 [15]. In contrast to *pals-22* mutants, we found that *pnp-1* mutants are slightly but significantly resistant to PA14 infection as compared to wild-type animals (Fig 4D). Because the NSY-1/SEK-1/ PMK-1 p38 MAP kinase pathway is one of the most important pathways for defense against *P. aeruginosa* in *C. elegans* [27–29] and we found it susceptible here in our studies (Fig 4E), we next investigated whether *pmk-1* was required for this increased resistance of *pnp-1* mutants (Fig 4E). Here we found that *pnp-1 pmk-1* double mutants had PA14 pathogen load similar to that of *pmk-1* single mutants, suggesting that *pmk-1* is required for the increased pathogen resistance of *pnp-1* mutants. We also found that loss of *pmk-1* in a *pals-22* mutant background further enhanced susceptibility in this background (Fig 4E).

Of note however, *pnp-1* mutants do not display increased survival when infected with *P. aeruginosa*, and in fact have slightly shorter lifespans compared to wild-type animals. Their lifespan is much longer than that of *pmk-1* mutants though, which is similar to *pnp-1 pmk-1* double mutants (S7 Fig). In addition, we found that expression of *pmk-1* regulated genes are much lower in *pnp-1 pmk-1* double mutants compared to wild-type animals, and are similar to that of *pmk-1* single mutants (S8 Fig). Altogether, these experiments analyzing gene expression, pathogen load and pathogen killing suggest that *pmk-1* acts downstream of, or in parallel to, both *pnp-1* and *pals-22* with respect to extracellular pathogen resistance.

During initial species description and characterization of *N. parisii*, we found very little role for *pmk-1* in resistance to *N. parisii* infection, as assessed by survival upon infection, and by meront and spore load quantified by Nomarski optics [10]. To analyze pathogen resistance in greater detail, we use the more recently developed method as described above to analyze *pmk-*

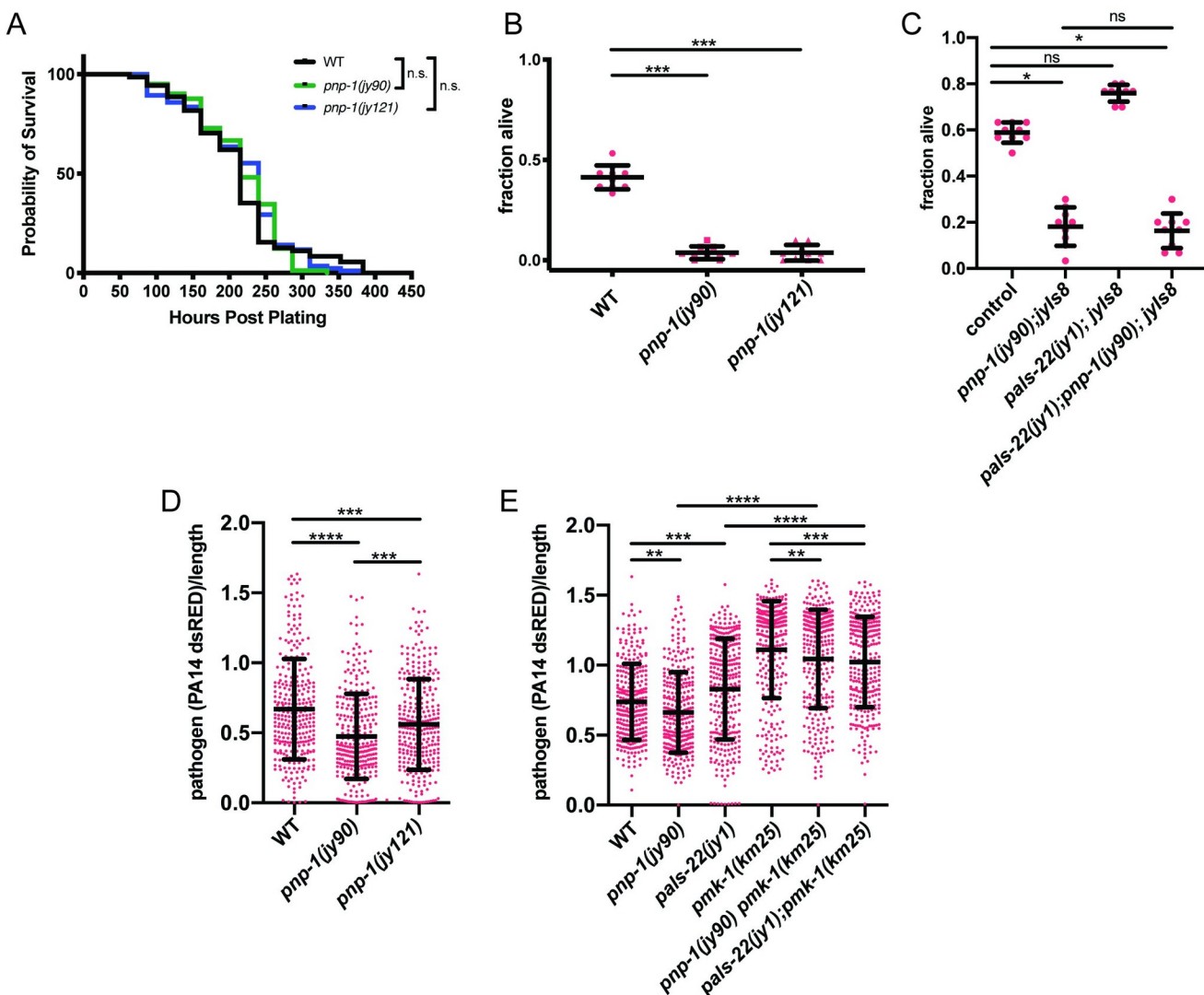

**Fig 4. Lifespan, thermotolerance and *P. aeruginosa* resistance phenotypes of *pnp-1* mutants.** A) Lifespan of wild-type animals, *pnp-1(jy90)* and *pnp-1 (jy121)* mutants. n = 90 animals per genotype. Results from one representative experiment of four independent experiments is shown (see S6 Fig for additional three experiments). Survival of each mutant population was compared to that of the wild-type population with the Log-rank (Mantel-Cox) test. B) Survival of wild-type animals, *pnp-1(jy90)* and *pnp-1(jy121)* mutants 24 hours after 2 hours 37˚C heat shock. C) Survival of wild-type animals, *pnp-1 (jy90)*, *pals-22(jy1)* and *pals-22(jy1); pnp-1(jy90)* mutants 24 hours after a 2 hour 37˚C heat shock. B, C) For one experiment, three plates per genotype with 30 worms per plate were tested. One dot represents the survival from one plate. The graphs show the mean survival of three independent experiments. Error bars are SD. D) Quantification of PA14-dsRED pathogen load in L4 stage wild-type animals, *pnp-1(jy90)* and *pnp-1(jy121)* mutants at 16 hpi. E) Quantification of PA14-dsRED pathogen load in L4 stage wild-type animals, *pnp-1(jy90)*, *pals-22(jy1)*, *pmk-1(km25)*, *pnp-1(jy90) pmk-1(km25)* and *pals-22 (jy1); pmk-1(km25)* mutants at 16 hpi. D, E) PA14-dsRED red fluorescence per animal was quantified with the COPAS Biosort machine and normalized to time-of-flight as proxy for the length of the animal. n = 300 animals per genotype. Graph shows the combined results of three independent experiments. Each dot represents an individual animal. Mean is shown with error bars as SD. B-E) **** indicates p < 0.0001 by the Kruskal-Wallis test.

*1* resistance at 3 hpi by counting sporoplasms. From this analysis we found that *pmk-1* mutants do have significantly enhanced susceptibility to *N. parisii* infection. Interestingly, this enhanced susceptibility can be suppressed by mutations in *pnp-1* or *pals-22* (S9A Fig). However, by feeding and quantifying fluorescent bead accumulation in the intestinal lumen, we found significantly decreased bead accumulation in *pnp-1 pmk-1* and *pals-22; pmk-1* double mutants compared to wild-type animals, indicating that the double mutants have less exposure to pathogen in the intestine (S9B Fig). Thus, the decreased susceptibility to *N. parisii* infection

observed in these double mutants may be due to decreased pathogen exposure. However, *pmk-1* mutant bead accumulation is not significantly different than that of wild-type animals indicating that their significantly enhanced susceptibility to *N. parisii* infection is not due to increased pathogen exposure.

Overall, these results demonstrate that while *pnp-1* and *pals-22* are both negative regulators of IPR gene expression and intracellular pathogen resistance, they have distinct phenotypes with respect to thermotolerance, lifespan and extracellular pathogen resistance.

### *pnp-1* mutants have increased expression of most IPR genes

Next, we sought to determine the full transcriptome changes in *pnp-1* mutants. Therefore, we performed RNA-seq analysis (S2 Table) on both *pnp-1* mutants as well as wild-type animals. By differential gene expression analysis (S3 Table), we determined that 326 and 294 genes are upregulated (p<0.05, no fold change cut off) in *pnp-1(jy90)* and *pnp-1(jy121)* mutants respectively, compared to wild-type animals. We found that 262 upregulated genes are common to both mutants and that this overlap is statistically significant (Fig 5A). In addition, we compared the genes upregulated in *pnp-1* and *pals-22* mutants and found a significant overlap between these gene sets (Fig 5B and S4 Table), as well as overlap with IPR genes (Fig 5C). Notably, of the 25 *pals* genes upregulated by *N. parisii* infection and Orsay virus, 22 are significantly up-regulated in both *pnp-1* mutant alleles (Fig 5C and 5D). In addition, several other genes that had previously been shown to be upregulated in the IPR are also upregulated in *pnp-1* mutants (Fig 5C and 5E and S4 Table).

To more globally evaluate the similarity between genes regulated by *pnp-1* and genes regulated by previously described IPR regulators, we performed Gene Set Enrichment Analysis (GSEA) (S5 and S6 Tables) [30], confirming similarity using hypergeometric testing (S7 Table). We determined that *pnp-1* significantly regulates genes that are upregulated by almost all known IPR triggers, including *N. parisii* infection, Orsay virus infection and treatment with the proteasome inhibitor bortezomib (Fig 5F). [Of note, *pnp-1* does not regulate the chitinase-like *chil* genes, which are induced by infection with the oomycete *Myzocytiopsis humicola*, a pathogen that can induce some, but not all IPR genes [17,31]. Moreover, *pnp-1* regulates genes that are also induced by ectopic expression of the RNA1 segment of Orsay virus that contains an active form of RNA-dependent RNA polymerase. These induced genes include many IPR genes [16].

### *pnp-1* negatively regulates genes involved in immunity to extracellular pathogens

Gene Ontology (GO) term analysis on the genes upregulated in *pnp-1* mutants showed statistically significant enrichment of genes involved in innate immune response (GO:0045087), carbohydrate binding (GO:0030246), defense response to Gram-negative bacteria (GO:0050829) and defense to Gram-positive bacteria (GO:0050830) (S8 Table) [32]. These GO terms were not previously found to be enriched in genes upregulated by *N. parisii* infection [13]. In addition, we performed a similar analysis with the Wormcat program that identifies significantly over-represented genes in a differentially expressed gene set and bins them into more refined categories than GO terms (Fig 5G and S9 Table) [33]. Wormcat analysis of *pnp-1* upregulated genes shows over-representation of genes involved in pathogen/stress response and proteasome proteolysis, which include genes that encode for proteins containing C-type lectin, CUB and F-box domains (Fig 5G). Some of these genes, such as *skr-3*, *fbxa-158* and *fxba-75* are also upregulated in *pals-22* mutants [17]. However, many non-IPR genes that are induced by bacterial infection, such as *irg-4*, *lys-1* and *dod-22*, are upregulated in *pnp-1* mutants, but not in

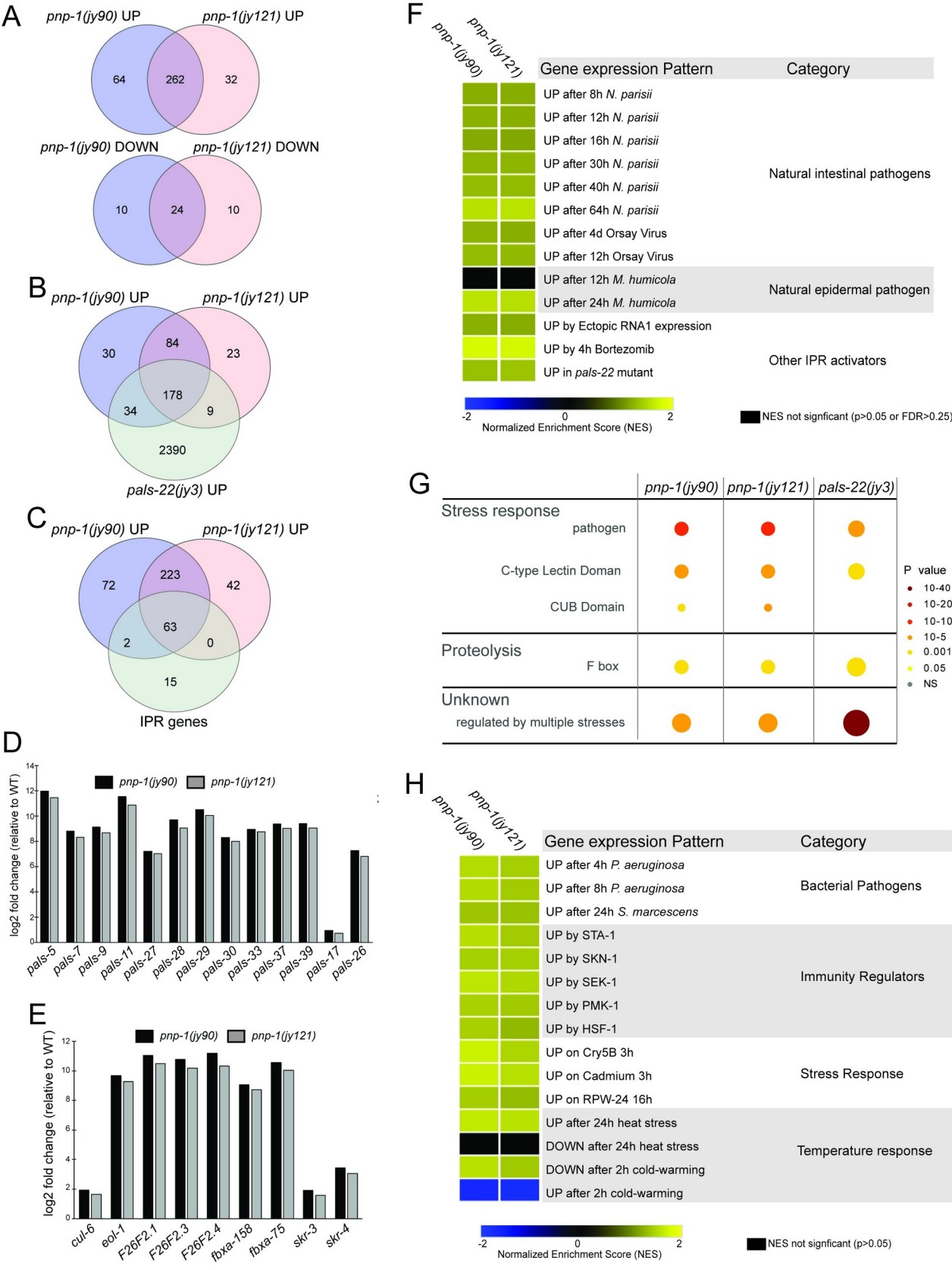

**Fig 5. RNA-seq analysis demonstrates that *pnp-1* represses expression of many IPR genes.** A) Venn diagram of differentially expressed genes in *pnp-1(jy90)* and *pnp-1(jy121)* mutants as compared to wild-type animals. Both up (rf = 31.5; p < 0.000e+00) and down (rf = 239.6; p < 3.489e-58) regulated genes have significant overlap between the two mutant alleles. B) Venn diagram of upregulated genes in *pnp-1(jy90)*, *pnp-1(jy121)* and *pals-22(jy3)* mutants as compared to wild-type animals. Upregulated genes in both *pnp-1(jy90)* (rf = 2.9; p < 4.566e-62) and *pnp-1(jy121)* (rf = 2.8; p < 1.591e-52) mutants have significant overlap with those upregulated in *pals-22(jy3)* mutants from a previous study [17]. C) Venn diagram of upregulated genes in *pnp-1(jy90)*, *pnp-1(jy121)* and the IPR genes (defined in [17]). Upregulated genes in both *pnp-1(jy90)* (rf = 28.8; p < 1.242e-88) and *pnp-1(jy121)* (rf = 30.9; p < 2.567e-87) mutants have significant with the IPR genes [17]. A-C) rf is the ratio of actual overlap to expected overlap where rf > 1 indicates overrepresentation and rf < 1 indicates underrepresentation (see S7 Table for more detail) D) Log2 fold-change of a subset of *pals* genes in *pnp-1(jy90)* and *pnp-1(jy121)* mutants normalized to wild-type. E) Log2 fold change of a subset of non-*pals* genes in *pnp-1(jy90)* and *pnp-1(jy121)* mutants normalized to wild-type. F) Correlation between genes differentially expressed by various known IPR activators and those differentially expressed in *pnp-1(jy90)* or *pnp-1(jy121)* mutants. G) Wormcat analysis for significantly enriched categories in differentially expressed gene sets of *pnp-1(jy90)* and *pnp-1(jy121)* mutants. Size of the circles indicates the number of the genes and color indicates value of significant over representation in each Wormcat category. H) Correlation of genes differentially expressed by bacterial pathogens, immune regulators, and various stressors to those differentially expressed in *pnp-1(jy90)* or *pnp-1(jy121)* mutants. F, H) Analysis was performed using GSEA 3.0 software, and correlations of genes sets (S5 Table) were quantified as a Normalized Enrichment Score (NES) (S6 Table). NES's presented in a heat map. Blue indicates significant correlation of downregulated genes in a *pnp-1* mutant with the tested gene set, yellow indicates significant correlation of upregulated genes in a *pnp-1* mutant with the tested gene set, while black indicates no significant correlation (p > 0.05 or False Discovery Rate < 0.25).

*pals-22* mutants [28]. This distinction may explain the contrast in resistance to bacterial pathogens between *pnp-1* and *pals-22* mutants.

As *pnp-1* mutants display resistance to PA14 and express various genes involved in bacterial defense, we used GSEA analysis and hypergeometric testing to determine the similarity between genes regulated by *pnp-1* and those regulated in response to infection by various bacterial pathogens (Fig 5H and S5–S7 Tables). We determined that genes upregulated in *pnp-1* mutants are significantly similar to those induced by the Gram-negative bacterial pathogens *P. aeruginosa* and *Serratia marcescens*. In addition, we used GSEA analysis to investigate the similarity between genes regulated by *pnp-1* and those regulated by known immune regulators. We determined that genes regulated by *pnp-1* are significantly similar to those regulated by the transcription factors *sta-1*, *skn-1*, *hsf-1*, as well as *pmk-1* and its upstream MAP Kinase Kinase *sek-1* (S10 Table). Previous GSEA analysis of genes up-regulated in *pals-22* mutants and by *N. parisii* infection showed very little similarity to genes induced by these extracellular pathogens and immune regulators. Taken together, these gene expression analyses indicate that, in addition to regulating the IPR, *pnp-1* may play a broader role in regulating *C. elegans* immunity to bacterial pathogens.

## Discussion

Here, we describe a role for purine metabolism in regulating intestinal epithelial cell defense against pathogen infection in the nematode *C. elegans* (Fig 6). Purine metabolism pathways are conserved from prokaryotes to humans and include the energy-expensive de novo synthesis pathway and the less energy-costly salvage pathway that recycles nucleotides (S3A Fig) [34]. While recent reports have implicated purine metabolism in *C. elegans* longevity and development, specific characterization of the salvage pathway has not previously been reported [35,36]. Through a forward genetic screen, we identified the salvage enzyme *pnp-1* as a negative regulator of the IPR, a common transcriptional response to intracellular pathogens. Our analysis of *pnp-1* represents the first characterization of purine nucleoside phosphorylase and, more broadly, the purine salvage pathway in *C. elegans*. We found that *pnp-1* mutants are resistant to the natural intracellular pathogens *N. parisii* and the Orsay virus (Fig 2). This resistance phenotype is in common with *pals-22* mutants, which also have upregulated IPR gene expression [17]. However, our analysis indicates that *pnp-1* acts in parallel to *pals-22/25* to regulate IPR gene expression (Figs S2B and 6A). Thus, *pnp-1* adds to a growing list of independent triggers of the IPR [13,16,17,31]. While much remains to be defined about the regulation

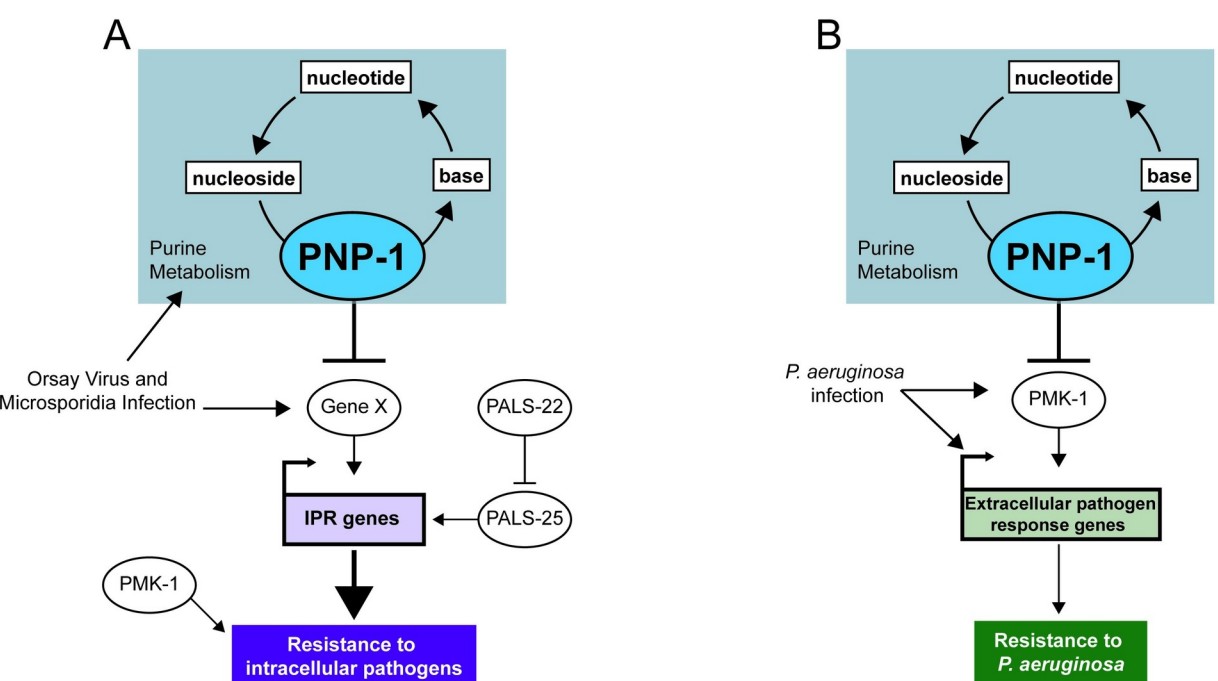

**Fig 6. Model for *pnp-1* regulation of immune responses.** A) *pnp-1* negatively regulates mRNA expression of IPR genes induced by infection with the intracellular pathogens the Orsay virus and *N. parisii* (microsporidia). Loss of *pnp-1* or *pals-22* results in constitutive expression of IPR genes and resistance to the Orsay virus and *N. parisii*. Epistasis analysis indicates that *pnp-1* acts in parallel to *pals-22/pals-25*. *pmk-1* (p38 MAPK) mutants display increased susceptibility to *N. parisii* as compared to wild-type animals. Because IPR genes are distinct from *pmk-1*-regulated genes, we favor a model where *pmk-1* acts in parallel to the IPR. B) *pnp-1* negatively regulates genes that are induced by various extracellular pathogens and *pnp-1* mutants are resistant to infection by the extracellular Gram-negative pathogen *P. aeruginosa*. Resistance to *P. aeruginosa* in *pnp-1* mutants requires *pmk-1*. In addition, upregulation of genes induced by wild-type *pmk-1* in *pnp-1* mutants requires functional *pmk-1*, suggesting that here, *pnp-1* functions upstream of *pmk-1*.

and outputs of IPR genes, these findings are consistent with the model that upregulation of IPR genes promotes defense against intracellular pathogens.

IPR gene upregulation in *pals-22* mutants has been associated with several other phenotypes, including increased proteostasis characterized by increased thermotolerance [15]. This increased thermotolerance of *pals-22* mutants is completely dependent on IPR genes that encode components of a multi-subunit cullin ring ubiquitin ligase complex, including the cullin *cul-6* and the F-box proteins *fbxa-75* and *fbxa-158* [20]. These IPR genes are also upregulated in *pnp-1* mutants, but surprisingly *pnp-1* mutants have greatly decreased thermotolerance compared to wild-type animals (Fig 4). One potential explanation for this distinction is that *pnp-1* mutants may have less metabolic flexibility upon heat shock than wild-type animals. *pnp-1* mutants are defective in the purine salvage pathway, which is less energy-costly than the de novo synthesis pathway. Under normal conditions the salvage pathway is preferentially used, but in response to high purine demands such as heat shock, the de novo pathway is activated to increase purine metabolic flux [37–43]. One possibility is that *pnp-1* mutants have constitutive use of the de novo pathway, and so are unable to increase purine metabolic flux upon heat shock, resulting in decreased thermotolerance. The observation that the *pnp-1* thermotolerance phenotype is epistatic to *pals-22* is consistent with this model, as there should be a requirement for appropriate purine metabolic flux upon heat shock, regardless of genetic background.

Another phenotype that is distinct between *pnp-1* and *pals-22* mutants is resistance to *P. aeruginosa*, with *pals-22* mutants having higher pathogen load and *pnp-1* mutants having

lower pathogen load of this extracellular bacterial pathogen (Fig 4). Transcriptomic analysis provided a likely explanation for this discrepancy, as *pnp-1* mutants have upregulated expression of many genes that are induced by bacterial infection and by previously described immunity pathways including the PMK-1 p38 MAPK pathway, whereas *pals-22* mutants do not. Consistent with the model that resistance of *pnp-1* mutants depends on *pmk-1*-induced genes, we find that *pmk-1* was required for the increased resistance of *pnp-1* mutants to *P. aeruginosa*, as well as the increased *P. aeruginosa* response gene expression in *pnp-1* mutants. However, it should be noted that *pmk-1* mutations can suppress the resistance phenotypes of other mutants, such as *daf-2* mutants, which have very few *pmk-1* genes upregulated [28]. Therefore, the requirement for *pmk-1* in the resistance of *pnp-1* mutants may be unrelated to their regulation of similar genes, although that is an attractive model (Fig 6B). Of note, *pnp-1* mutants did not have increased survival upon *P. aeruginosa* infection compared to wild-type animals, despite their lower pathogen load. Perhaps this lowered tolerance of infection reflects less metabolic flexibility in these mutants upon bacterial pathogen infection, similar to their lower tolerance of thermal stressors.

How does constitutive upregulation of IPR genes in both *pnp-1* and *pals-22* mutants lead to increased resistance to intracellular pathogens at early timepoints? The ubiquitin ligase components mentioned above appear to play a minor role in pathogen resistance [13,20], indicating that there is either extensive redundancy in these components, other IPR genes are more important, or there is a non-transcriptional component that enables resistance in *pnp-1* and *pals-22* mutants. Regardless of which genes mediate the increased resistance in these mutants, it should be noted that the *N. parisii* parasite cells at 3 hpi are likely the result of initial invasion events, so some host factor regulating intestinal cell invasion may be altered in these mutants. Further analysis of individual IPR genes may yield insight into the mechanism of resistance against *N. parisii* in these mutants.

Which purine metabolites regulate the expression of IPR and other pathogen response genes in *pnp-1* mutants? Our metabolomics analysis confirmed that, as predicted, *pnp-1* mutants have increased levels of inosine, the nucleoside substrate for *pnp-1*, and decreased levels of hypoxanthine, the purine base product of *pnp-1* (Fig 1). No other metabolites were found to have altered levels in these mutants (S3 Fig). Therefore, inosine may be an activator, or hypoxanthine a repressor, of pathogen response gene expression, although we saw no effect on *pals-5p::gfp* expression when adding these metabolites to worms. It is possible that altered levels of some other unknown metabolites in *pnp-1* mutants are regulators of the IPR, or perhaps *pnp-1* regulates the IPR in a manner independent of its enzymatic activity. Support for the model that the catalytic activity of *pnp-1* is required for its effects on the IPR comes from the *pnp-1(jy90)* allele, which has a conserved serine mutated to leucine. When this residue is mutated in the human ortholog it leads to an inactive enzyme, suggesting that *pnp-1* enzymatic function is key to its regulation of the IPR [24].

All life, including eukaryotes, prokaryotes and viruses, require purines as part of deoxynucleotides for DNA and ribonucleotides for RNA. Obligate intracellular pathogens as diverse as viruses and the eukaryotic pathogens *Plasmodium falciparum* and microsporidia depend on their hosts for purines, with several types of transporters identified in eukaryotic pathogens used to steal purines from hosts [4–6,44–46]. Extensive work on purine metabolism and its effects on human physiology have come from studies of so-called 'inborn errors of metabolism', which are often due to mutations in purine enzymes [21,38,47–50]. Interestingly, although mutations in enzymes of the purine de novo and salvage pathways cause various disorders (e.g. deafness, intellectual disability, motor dysfunction, renal failure), only mutations in purine salvage enzymes have been reported to result in immunodeficiency due to T-cell dysfunction [51,52]. While the focus is on the detrimental effects of these mutations, it is

interesting to speculate that there may be some advantage conferred by these mutations in terms of resistance to infection, perhaps in the heterozygote state, or against certain pathogens and/or in certain cell types. While mutations in human PNP cause severe combined immuno-deficiency, this phenotype appears to be due to apoptosis of T cells [53,54]. Less has been described about the role of PNP in intestinal epithelial cells, which is the cell type where *C. elegans pnp-1* mutants have increased pathogen resistance. Further work on purine metabolism and how it regulates immunity may help shed light on whether altered levels of purine metabolites may be sensed to enable hosts to monitor the effects of pathogen infection as part of surveillance immunity to induce epithelial cell defense.

## Methods

### *C. elegans* strains

All worm strains were maintained by standard methods [55]. Briefly, worms were grown on NGM plates seeded with *E. coli* strain OP50-1 and grown at 20˚C. All mutant and transgenic strains were backcrossed a minimum of three times. See S1 Table for a list of strains used. For several experiments, synchronized L1 worms were prepared by bleaching gravid adults to isolate embryos that hatched in M9 buffer into starved, synchronized L1 worms.

### Forward mutagenesis screening and cloning of *pnp-1(jy90)*

Ethyl methane sulfonate (EMS) (Sigma) mutagenesis of *jyIs8* [*pals-5p*::*gfp*, *myo-2p*::*mCherry*] animals was performed by standard procedures [56]. P0 worms were incubated with 50 mM EMS at 20˚C for 4 hours with constant rotation. Using a fluorescence dissecting microscope (Zeiss Discovery V8), mutant F2 animals ectopically expressing *pals-5p*::*gfp* were isolated. ~26,000 haploid genomes were screened. From this screen, we identified 9 mutant alleles that result in robust ectopic GFP expression. Four mutants failed to complement the constitutive *pals-5p*::*gfp* expression phenotype of *pals-22* mutants, indicating that they likely have mutations in *pals-22*. The other five alleles complement each other for the *pals-5p*::*gfp* expression phenotype, indicating that they have mutations in 5 distinct genes. *jy90* was mapped to Linkage Group (LG) IV using visible makers contained in the strains ERT507, ERT508, and ERT509 and confirmed by linkage group mapping using SNP primers [57]. DNA was prepared using Puregene Core kit (QIAGEN) for whole genome sequencing and submitted to Beijing Genomics Institute for sequencing with a 100 bp paired-end Illumina HiSeq 4000 with 30X coverage. Analysis identified one gene (*pnp-1*) on LG IV that harbored a variant predicted to alter gene function. *pnp-1(jy90)* contains a G to A substitution that should convert serine 51 to leucine in isoform A and serine 68 to leucine in isoform B.

### CRISPR/Cas9-mediated gene deletion of *pnp-1*

A co-CRISPR protocol was used to generate a complete deletion of *pnp-1* [58]. Two CRISPR RNAs (crRNA) were designed to target the 5' end (TGATTTCATTGGCTTCCACG) and 3' end (AGTTTTTTCTGTGAACCACG) of the gene. A crRNA against *dpy-10* was used as a control. All three cRNAs and tracrRNA were synthesized by Integrated DNA Technologies (IDT) and resuspended in IDT nuclease-free duplex buffer to 100 μM. An injection mix was made by first annealing 0.5 μl of three crRNAs with 2.5 μl tracrRNA, then complexing the annealed sgRNA with purified 3.5 μl of 40 μM Cas9 protein. This mix was injected into the gonads of *jyIs8* worms. Dumpy (Dpy) and/or GFP-positive F1 animals were selected and genotyped for deletions of the *pnp-1* locus. After submitting to Sanger sequencing to confirm the presence of

a complete deletion, one strain was selected and named allele *jy121*. *pnp-1(jy121)* was back-crossed to *jyIs8* three times and then outcrossed to N2.

## Generation of transgenic strains

**For the transgenic expression reporter of PNP-1 protein.** The *pnp-1* TransgeneOme fosmid (K02D7.1[20219]::S0001_pR6K_Amp_2xTY1ce_EGFP_FRT_rpsl_neo_FRT_3xFlag) dFRT::unc-119-Nat) was injected into EG6699 at 100 ng/μl to generate strain ERT879.

For fosmid rescue of the *pnp-1(jy90)* mutant phenotype:

The following injection mix was made and injected into *pnp-1(jy90)* mutants to generate strain ERT869: *myo-2p*::*mCherry* (10 ng/μl), *pnp-1* transgeneOme fosmid (25 ng/μl), genomic N2 DNA (65 ng/μl).

**For single-copy tissue-specific *pnp-1* rescue strains.** CRISPR/Cas9 genome editing was used to insert tissue-specific expression cassettes at cxTi10882 on Chromosome IV as previously described, in a so-called "Cas-SCI (Single-Copy Insertion) technique" [59]. Briefly, we generated plasmids pET721 (*vha-6*::*pnp-1*::3XFLAG::GFP::*unc-54*) and pET720 (*unc-119*::*pnp-1*::3XFLAG::*unc-54*) by assembling the tissue-specific cassettes (including *unc-54* 3' UTR) into plasmid pCZGY2729 such that the tissue-specific cassettes and the hygromycin resistance gene were flanked by homology arms to cxTi10882. These plasmids (25 ng/μl) were then injected into N2 animals along with pCZGY2750 that expresses Cas9 and sgRNA for cxTi10882. pCFJ10 *myo-3p*::*mCherry* (5 ng/μl) and pCFJ90 *myo-2p*::*mCherry* (2 ng/μl) were used a co-injection markers. Genomic insertion was determined by identifying animals resistant to hygromycin that did not express the co-injection markers. Single-copy insertion lines were verified by genotyping.

## RNA extraction and qRT-PCR

RNA was extracted using TRI reagent and 1-Bromo-3-chloropropane (Molecular Research Center, Inc.), according to the manufacturer's instructions, and converted to cDNA using the iScript (Bio-Rad) cDNA synthesis kit. qRT-PCR was performed using iQ SYBR green super-mix (Bio-Rad) and various gene specific primers (S1 Table) on a BioRad CFX Connect real-time system. Each biological replicate was analyzed in technical duplicates and normalized to *nhr-23 or snb-1*, control genes that did not change expression in these experiments. Three experimental replicates were performed with one biological replicate for each condition.

## Thermotolerance assays

Gravid adults were picked to NGM plates and grown at 20˚C. L4 progeny from these adults were picked to fresh NGM plates and submitted to a heat shock of 37˚C for two hours. Animals were allowed to recover for 30 minutes at room temperature, then incubated at 20˚C for 24 hours, and then scored for survival. During both the heat shock and recovery, plates were placed in a single layer (plates were not stacked on top of each other). Worms were defined as dead by lack of movement on the plate, lack of pharyngeal pumping and lack of response to touch. For each replicate, three plates containing 30 worms were scored per genotype. Three experimental replicates were performed.

## Targeted metabolomics

Synchronized L1 worms were plated on NGM plates seeded with OP50-1 and grown to Day 1 adult stage at 20˚C. Fifteen 6 cm plates were used for each genotype per experiment and 6 experiments were performed. Metabolite extraction and LC-MS were performed with the

Penn State Metabolomics Core facility as previously described [60]. Raw Data were processed with MS-DIAL and metabolite levels were corrected to chlorpropamide, an internal standard. Selected metabolites were identified by m/z and column retention time values of known standards. Normalized area under the curve for each metabolite is represented as arbitrary units in graphs.

### Dietary complementation of inosine and hypoxanthine

NGM plates containing cell culture grade inosine or hypoxanthine were made by adding the chemicals (purchased from Sigma) to hot liquid NGM before pouring. NGM plates with varying concentrations of hypoxanthine were made and seeded with 10X concentration of OP50-1. Synchronized L1s of *pnp-1(jy90); jyIs8* were plated and *pals-5*::*gfp* expression was tracked every 24 hours for 4 days. Hypoxanthine was tested at 0 mM, 10 mM and 100 mM in one experiment, and at 0 mM, 25 mM and 100 mM in a second experiment.

NGM plates with 0 mM, 19 mM and 79 mM inosine were made and seeded with 10X concentration of OP50-1. Developmental delay was observed when synchronized *jyIs8* L1s were grown on NGM-inosine plates. Therefore, synchronized *jyIs8* L1s were grown to L4 on NGM plates and then shifted to NGM-inosine plates. *pals-5p*::*gfp* expression was assessed at 24 hours and 48 hours after exposure. Two independent experimental replicates were performed.

For all supplementation experiments, 50 animals were scored for *pals-5p*::*gfp* expression at each timepoint in each experiment.

### *N. parisii* infection

*N. parisii* spores were isolated as previously described [25]. 1200 synchronized L1 worms were mixed with $5 \times 10^6$ *N. parisii* spores, 25 μl 10X concentrated OP50-1 bacteria and M9 to bring the total volume to 300 μl. This mixture was then plated on room temperature unseeded 6 cm NGM plates, allowed to dry and then incubated at 25˚C for 3 hours or 30 hours. Three plates were used per genotype. Animals were fixed in 4% paraformaldehyde and then stained using a FISH probe specific to *N. parisii* ribosomal RNA conjugated to Cal Fluor 610 dye (Biosearch Technologies). For the 3 hpi timepoint, pathogen load was determined by counting sporoplasms per worms using 40x objective on a Zeiss AxioImager M1 microscope. For each replicate, 75 animals per genotype were quantified. Three experimental replicates were performed. For the 30 hpi timepoint, pathogen load was quantified using the COPAS Biosort machine (Union Biometrica). The *N. parisii* FISH signal for each worm was normalized to the length of the worm using time-of-flight measurements. For each replicate, 100 animals per genotype were quantified.

### Bead feeding assay

1200 synchronized L1 worms were mixed with 6 μl fluorescent beads (Fluoresbrite Polychromatic Red Microspheres, Polysciences Inc.). 25 μl 10X concentrated OP50-1 bacteria and M9 to bring the total volume to 300 ul. This mixture was then plated on room temperature unseeded 6 cm plates, allowed to dry for 5 mins and then incubated at 25˚C for 5 mins. Plates were immediately shifted to ice, washed with ice cold PBS-Tween and fixed in 4% paraformaldehyde. Worms were imaged with the 4x objective on an ImageXpress Nano plate reader. Using FIJI software, the integrated density of bead fluorescence per worm was quantified from which background fluorescence was subtracted giving the Corrected Total Fluorescence for each worm. For each replicate, 50 animals per genotype were quantified. Three experimental replicates were performed.

## *P. aeruginosa* infection

Slow Killing (SK) plates with 50 μg/ml ampicillin were seeded with overnight cultures of PA14-dsRED [61], and then incubated at 37°C for 24 hours followed by a 24 hour incubation at 25°C. 3000 synchronized L1 worms were plated onto NGM plates seeded with OP50-1 and allowed to grow to L4 at 20°C. L4 worms were washed with M9, transferred to the PA14 dsRED SK plates and incubated at 25°C for 16 hours. Worms were then washed with M9 and PA14-dsRED fluorescence per animal was quantified using the COPAS Biosort machine (Union Biometrica). For each replicate, 100 animals per genotype were quantified. Three experimental replicates were performed.

## Orsay virus infection

Orsay virus filtrates were prepared as previously described [13]. ~2,000 synchronized L1s were plated onto one 10 cm NGM plate containing a lawn of OP50-1 *E. coli* per worm strain. Plates were incubated at 20°C for ~44 hours until the L4 stage. 30 μl of the Orsay virus filtrate, after dilution by a factor of 10 in M9, was mixed with 150 μl of a 10X concentration of OP50-1 *E. coli* and 600 μl M9 buffer. 780 μl of this final mixture was added to the L4 animals on 10 cm plates, dried in the Laminar flow hood at room temperature, and incubated at 20°C for 24 hours. RNA was extracted isolated using Tri-reagent (Molecular Research Center, Inc) and converted to cDNA with iScript (Bio-Rad) cDNA synthesis kit. qRT-PCR was performed using iQ SYBR green supermix (Bio-Rad) and primers specific to RNA1 of Orsay virus. All gene expression was normalized to *snb-1* expression, which does not change upon conditions tested. Three experimental replicates were performed.

## Lifespan and killing assays

For lifespan assays, ~50 synchronized L1 worms/plate per strain were plated onto six 3.5 cm tissue culture-treated NGM plates seeded with OP50-1. Plates were incubated at 25°C. After 66 hours, 30 adults per strain were scored in triplicate and live animals were transferred to fresh plates. Plates were incubated at 25°C and live worms were transferred every 24 hours until death or until progeny production stopped. Survival was measured every 24 hours and worms that did not respond to touch were scored as dead. Animals that died from internal hatching or crawled off the plate were censored. Four experimental replicates were performed.

For *N. parisii* killing assays, ~100 synchronized L1 worms were plated with a mixture of 50 μl of a 10X concentration of OP50-1 *E. coli* and 5 x 10⁴ *N. parisii* spores onto a 3.5 cm tissue culture-treated NGM plate. Number of spores added was determined by finding the dosage that resulted in approximately 50% killing rate in wild-type worms after 100 hours. Three experimental replicates were performed. *P. aeruginosa* (PA14) killing assays were performed as previously described [62]. Three experimental replicates were performed.

## RNA-seq sample preparation and sequencing

~3,000 synchronized L1 worms/plate per strain were plated onto a 10 cm NGM plate seeded with OP50-1 and allowed to grow for 56 hours at 20°C. As *pnp-1* mutants grow slightly slower, wild-type L1 worms were plated 1 hour after *pnp-1(jy90)* and *pnp-1(jy121)* so they were synchronized in age at harvest time. RNA was extracted using TRI reagent and BCP (Molecular Research Center, Inc.) according to the manufacturer's instructions and additionally purified using the RNeasy cleanup kit with gDNA Eliminator spin columns (Qiagen). RNA quality was assessed using a TapeStation system. Sequencing libraries were constructed using TruSeq stranded mRNA method and sequenced using run type PE100 on an Illumina NovaSeq6000

sequencer (Illumina). RNA quality assessment and RNA-seq were conducted at the IGM Genomics Center, University of California, San Diego, La Jolla, CA. RNA-seq reads were uploaded to the NCBI GEO database with Accession number GSE165786.

### RNA-seq and functional expression analysis

In Rstudio, sequencing reads for *pnp-1(jy90)*, *pnp-1(jy121)* and N2 were aligned to the Wormbase WS235 release using Rsubread and quantified using Featurecounts. Using the Galaxy web platform and the public server at usegalaxy.org, differential gene expression analysis was performed using limma-voom in which undetected and lowly expressed genes (CPM of less than one) were filtered out. An adjusted p-value of 0.05 and no fold-change cutoff was used to define differentially expressed genes. This gene set was used for GO term enrichment analysis (Galaxy) [63] and Wormcat analysis (wormcat.com) [33].

Gene Set Enrichment Analysis v3.0 software was used for functional analysis [30]. Normalized RNA-sequence expression data was converted into a GSEA compatible filetype and used for analysis. The gene sets for comparison were made in Excel and then converted into a GSEA compatible file type. Independent GSEA analysis was performed for *pnp-1(jy90)* vs N2 and *pnp-1(jy121)* vs N2 gene sets. For both analyses, a signal-to-noise metric with 1000 permutations was used. Heatmaps for the NES results were made using Morpheus (https://software.broadinstitute.org/morpheus/). Gene sets that showed significant similarity to *pnp-1(jy90)* and *pnp-1(jy121)* differentially expressed genes were submitted to a hypergeometric test (nemates.org). Representation factors and their p-values for the overlap of *pnp-1(jy90)* or *pnp-1(jy121)* and individual gene sets were calculated using the size of the RNA-seq data set after filtering [11539 for *pnp-1(jy90)* and 11537 for *pnp-1(jy121)*].

### Statistics

**For all data.** Statistical analysis was performed in Prism 8. Means with error bars as standard deviation are presented unless otherwise noted. For sporoplasm experiments, the box represents the 50% of the data closest to the median and the whiskers span the values outside the box. For metabolite experiments, error bars represent the standard error of the means. Statistical significance indicated as follows: ns indicates not significant, $^*$ indicates $p < 0.05$, $^{**}$ indicates $p < 0.01$, $^{***}$ indicates $p < 0.001$, $^{****}$ indicates $p < 0.0001$.

**For lifespan and survival assays.** The survival of each mutant population was compared to that of the wild-type population in Prism 8 with the Log-rank (Mantel-Cox) test. A *p*-value $<0.05$ was considered significantly different from control.

**For qPCR assays.** Unpaired, one-tailed student t-test was used. A *p*-value $< 0.05$ was considered significantly different.

**For all pathogen load and thermotolerance assays.** For comparisons between three or more genotypes, mean ranks for three biological replicates were compared using the Kruskal-Wallis test with the Dunn's correction. For comparisons between two genotypes, means of pathogen loads for three biological replicates were compared using a Mann Whitney test. A *p*-value $< 0.05$ was considered significantly different.

**For RNA-seq and Functional Expression analysis.** For GSEA, a *p*-value $< 0.05$ or FDR $< 0.25$ was considered significantly similar. For the hypergeometric test, a *p*-value $< 0.05$ was considered significantly similar.

### Imaging

Worms were anesthetized with 10 mM sodium azide and mounted on 2% agarose pads for analysis on a Zeiss Axioimager Z1 compound microscope.

## Supporting information

**S1 Fig. Alignment of PNP proteins across species.** Alignment of PNP protein sequences from *C. elegans* (two isoforms, PNP-1a and PNP-1b), *Drosophila melanogaster* (DmPNP), *Mus musculus* (MsPNP) and *Homo sapiens* (HsPNP). Similar amino acids shaded gray, identical amino acids shaded black. Red indicates the serine converted to leucine in *pnp-1 (jy90)* mutants. Clustal Omega was used to perform the alignment (https://www.ebi.ac.uk/Tools/msa/clustalo/). BoxShade (https://embnet.vital-it.ch/software/BOX_form.html) was used to annotate sequence homology.
(TIFF)

**S2 Fig. Quantification of IPR gene expression in *pals-22*, *pnp-1* and *pals-25*; *pnp-1* mutants.** A) qRT-PCR of a subset of IPR genes in *pals-22(jy1)* and *pnp-1(jy90)* mutants. Synchronized animals grown for 44 hours at 20˚C post L1 were used. Graph shows the mean fold change of four independent experiments. B) qRT-PCR of a subset of IPR genes in *pnp-1(jy90)* and *pals-25(jy81); pnp-1(jy90)* mutants. Mixed stage populations of animals were used. Graph shows the mean fold change of two independent experiments. A, B) Fold change in gene expression is shown relative to control animals. Error bars are standard deviation (SD). Red dots indicate values from individual experiments *** indicates $p < 0.001$ by a one-tailed t-test.
(TIF)

**S3 Fig. Quantification by LC-MS of purine metabolites in *pnp-1* and *pals-22* mutants.** A) Schematic of purine synthesis pathways with select metabolites included. The salvage pathway is highlighted in red. The de novo pathway and metabolites are highlighted in blue. Metabolites that are not highlighted are common to both pathways. B-D) Quantification of adenosine, de novo specific metabolites and purine nucleotides, respectively (inosine and hypoxanthine are shown in Fig 1F). Graphs show the mean amount (in log10 scale) of the metabolites of six independent experiments for *pnp-1(jy121)*, *pnp-1(jy90)* and *pals-22(jy1)* mutants and five independent experiments for wild-type animals. Dots (red or black) show individual values for each experiment. Error bars are SEM. Unless otherwise indicated, there is no significant difference in metabolite amounts in *pnp-1* mutants or *pals-22* mutants compared to control as determined by the Kruskal-Wallis test. ** indicates $p < 0.01$ by the Kruskal-Wallis test. Abbreviations used: R5P is ribose-5-phospate; GAR is glycineamide ribonucleotide; CAIR is 5'-phosphoribosyl-4-carboxy-5-aminoimidazole; SAICAR is succinylaminoimidazole carboxamide ribotide; AICAR is 5-Aminoimidazole-4-carboxamide ribonucleotide; IMP is inosine monophosphate; XMP is xanthine monophosphate; GMP is guanosine monophosphate; S-AMP is adenylosuccinate; AMP is adenosine monophosphate.
(TIF)

**S4 Fig. Individual experiments for survival of wild-type, *pnp-1(jy90)* and *pnp-1(jy121)* after *N. parisii* infection.** n = 120 per genotype for each replicate. **** indicates $p < 0.0001$ by the Log-rank (Mantel-Cox) test.
(TIF)

**S5 Fig. DiI filling of animals expressing *pnp-1::EGFP::3XFLAG*.** Transgenic *pnp-1::egfp::3Xflag* TransgeneOme animals, stained with the lipophilic fluorescent dye DiI, which labels a subset of amphid neurons in red. GFP-expressing cells (indicated by green arrows) are distinct from DiI-labled cells (indicated by red arrows), suggesting that *pnp-1* is not expressed in the subset of amphid neurons that take up DiI. Each row is an individual animal and scale bar is 20 μm. The head of the animal is shown with anterior to the left. Worm bodies are outlined in white. Asterisks indicate GFP-expressing intestines. The exposure time for GFP in the bottom

row is higher than that of the above two panels to better visualize the GFP-expressing processes extending anteriorly.
(TIF)

**S6 Fig. Individual experiments for lifespan of wild-type animals, *pnp-1(jy90)* and *pnp-1 (jy121)* mutants.** n = 90 per genotype for each replicate. **** indicates p < 0.0001 by the Log-rank (Mantel-Cox) test.
(TIF)

**S7 Fig. Individual experiments for survival of wild-type, *pnp-1(jy90), pmk-1(km25)* and *pnp-1(jy90) pmk-1(km25)* after *P. aeruginosa* infection.** n = 90 per genotype for each replicate. **** indicates p < 0.0001 by the Log-rank (Mantel-Cox) test.
(TIF)

**S8 Fig. Quantification of *pmk-1* regulated gene expression in the *pmk-1* and *pnp-1(jy90) pmk-1(km25)*.** A) qRT-PCR of a subset of *pmk-1* regulated genes in *pnp-1(jy90), pmk-1(km25) and pnp-1(jy90) pmk-1(km25)*. Synchronized animals grown for 44 hours at 20˚C post L1 were used. B) qRT-PCR of a subset of *pmk-1* regulated genes in *pnp-1(jy90), pmk-1(km25) and pnp-1(jy90) pmk-1(km25)*. Synchronized animals grown for 56 hours at 20˚C post L1 were used. A, B) Fold change in gene expression is shown relative to control. Graphs show the combined results of three independent experiments. Red dots indicate values from individual experiments. **** indicates p < 0.0001 by a one-tailed t-test.
(TIF)

**S9 Fig. Quantification of *N. parisii* sporoplasm number and fluorescent bead accumulation in *pmk-1, pnp-1(jy90) pmk-1(km25)* and *pals-22(jy1); pmk-1(km25)* mutants.** A) Quantification of *N. parisii* sporoplasm number in wild-type animals, *pnp-1(jy90), pals-22(jy1), pmk-1 (km25), pnp-1(jy90) pmk-1(km25)* and *pals-22(jy1); pmk-1(km25)* mutants at 3 hpi. n = 400 animals per genotype. The box represents the 50% of the data closest to the median while the whiskers span the values outside the box. Graph shows combined results of four independent experiments. B) Quantification of fluorescent bead accumulation in wild-type animals, *pmk-1 (km25), pnp-1(jy90) pmk-1(km25), pals-22(jy1); pmk-1(km25)*, and *eat-2(ad465)* mutants. n = 150 animals per genotype. The corrected total fluorescence per worm was calculated and normalized to worm area. Each dot represents an individual animal. Graph shows combined results of three independent experiments. A, B) **** indicates p < 0.0001 by the Kruskal-Wallis test.
(TIF)

**S1 Table. Lists of strains, plasmids and primers.**
(XLSX)

**S2 Table. RNA-seq statistics.**
(XLSX)

**S3 Table. Differentially expressed genes lists.**
(XLSX)

**S4 Table. Detailed list of overlaps of upregulated genes in *pnp-1* mutants and those upregulated in *pals-22* mutants or the IPR gene set.**
(XLSX)

**S5 Table. Gene sets used for GSEA.**
(XLSX)

**S6 Table. Detailed GSEA results.**
(XLSX)

**S7 Table. Detailed results of hypergeometric testing.**
(XLSX)

**S8 Table. Detailed GO Term results.**
(XLSX)

**S9 Table. Detailed Wormcat results.**
(XLSX)

**S10 Table. Detailed list of overlaps of upregulated genes in *pnp-1* mutants and those upregulated by PA14 infection, in *pmk-1* mutants or in *sek-1* mutants.**
(XLSX)

# Acknowledgments

We thank Spencer Gang and Vladimir Lazetic for helpful feedback on the manuscript. We thank Jessica Sowa for preparing the viral filtrate, and Yishi Jin and Matt Andrusiak for reagents and help with the CRISPR/Cas9-mediated single-copy insertion "Cas-SCI" technique. This publication includes data generated at the UC San Diego IGM Genomics Center utilizing an Illumina NovaSeq 6000. Some strains were provided by the CGC.

# Author Contributions

**Conceptualization:** Eillen Tecle, Emily R. Troemel.

**Data curation:** Eillen Tecle, Latisha Franklin, Ryan S. Underwood.

**Formal analysis:** Eillen Tecle, Crystal B. Chhan, Latisha Franklin, Ryan S. Underwood.

**Funding acquisition:** Eillen Tecle, Wendy Hanna-Rose, Emily R. Troemel.

**Investigation:** Eillen Tecle, Crystal B. Chhan, Latisha Franklin.

**Supervision:** Wendy Hanna-Rose, Emily R. Troemel.

**Visualization:** Eillen Tecle.

**Writing – original draft:** Eillen Tecle, Emily R. Troemel.

**Writing – review & editing:** Eillen Tecle, Crystal B. Chhan, Latisha Franklin, Ryan S. Underwood, Wendy Hanna-Rose, Emily R. Troemel.

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
