## [Decision Letter · Decision Letter 0]

25 Feb 2021

Dear Emily,

Your manuscript was reviewed by three experts in our field and at the editorial level. The reviewers and editors were uniformly enthusiastic about this manuscript, and feel that it will likely be appropriate for publication in PLOS PATHOGENS. The reviewers raised a few issues that I hope you can address in a revised version of the manuscript. In particular, I agree with reviewer 2 that hypoxanthine supplementation studies would add mechanistic detail to your study, although I do not view such an experiment as a prerequisite for publication. I have entered a decision of "Minor Revision," so in formulating your revised manuscript, I encourage you to only conduct new studies that could be preformed in a time-efficient manner.

Sincerely,

Read Pukkila-Worley, M.D.

Guest Editor

PLOS Pathogens

James Collins III

Section Editor

PLOS Pathogens

Kasturi Haldar

Editor-in-Chief

PLOS Pathogens

orcid.org/0000-0001-5065-158X

Michael Malim

Editor-in-Chief

PLOS Pathogens

orcid.org/0000-0002-7699-2064

Reviewer Comments (if any, and for reference):

Reviewer's Responses to Questions

**Part I - Summary**

Reviewer #1: This study presents a novel regulator of the C. elegans intracellular pathogen response (IPR), which is the common transcriptional response to intracellular pathogens such as microsporidia and viruses. Using a previously established IPR pals-5p::GFP reporter strain in a forward genetic screen, the authors identify the purine nucleoside phosphorylase gene pnp-1 as negative regulator of the IPR. The function of pnp-1 in regulating IPR gene expression was then confirmed with a CRISPR/Cas9 deletion allele. Liquid Chromatography-Mass Spectrometry metabolomic analysis of the pnp-1 mutants was carried out, revealing that pnp-1 indeed functions as purine nucleoside phosphorylase in C. elegans, converting purine nucleosides into free purine bases. pnp-1 mutants display reduced intracellular pathogen load and increased survival upon infection with the intracellular eukaryotic pathogen Nematocida parisii. The authors show that pnp-1 functions in the intestine to regulate IPR gene expression and resistance. pnp-1 mutants also display reduced pathogen load upon infection with the extracellular pathogen Pseudomonas aeruginosa and transcriptomic analysis of pnp-1 mutants reveals that pnp-1 regulated genes are similar to genes regulated by known regulators of pathogen defenses and genes induced by infection with Gram-negative pathogens, suggesting contributions of pnp-1 to resistance to both intracellular and extracellular pathogens.

The findings presented in this manuscript are significant and of broad interest. The authors identify pnp-1 as novel regulator of the C. elegans IPR and thoroughly analyze its role, also in context with other known IPR regulators (namely pals-22). Moreover, they provide first evidence of a link between purine metabolism and the regulation of C. elegans pathogen defenses to both intracellular and extracellular pathogens. Although the mechanistic basis remains unclear, the findings are novel and valuable for the field and will certainly spark further studies. One could ask for more experiments to confirm the role of pnp-1 and purine metabolism in regulating C. elegans defenses to extracellular and intracellular pathogens (e.g analysis of other purine biosynthesis pathway mutants), but publication of the presented results as they stand is justified. The manuscript is well written and the experiments, which were done in a convincing and thorough manner, are well presented.

Reviewer #2: This manuscript identifies a new player in the C. elegans Intracellular Pathogen Response (IPR) that defends against viruses and fungal-like microsporidia, namely the purine nucleoside phosphorylase pnp-1, which is involved in the purine salvage pathway. Using a mutant from an unbiased forward genetics screen and a CRISPR-Cas9 generated null mutant, the authors show that loss of pnp-1 activates many genes of the IPR. This activation correlates with increased resistance to intracellular and, surprisingly and unlike mutants in other negative regulators of the IPR, extracellular pathogens. Targeted metabolomics show that C. elegans pnp-1 is a bona fide purine nucleoside phosphorylase whose loss reduces hypoxanthine and increases inosine levels. Genetic rescue indicates that pnp-1 acts in the intestine. RNA-seq studies show that loss of pnp-1 activates a response that resembles that of another IPR mutants, pals-22, but also includes a separate transcriptional program that likely drives resistance to extracellular pathogens and overlaps with classical pathways therein (pmk-1 etc).

The genetic (two mutants, used almost in all assays!), genomic (also done in both mutants!), and metabolomic analysis is very well done and well analyzed and described; props to the authors for such a nice dataset. The paper is also well composed: easy to read, the figures are clear, and the conclusions are well supported by the data. The information presented is to my knowledge new and provides an interesting new insight into the roles a highly conserved enzyme plays in the intra- and extracellular pathogen response.

The precise role of pnp-1 and its substrate and product remain somewhat cryptic, however. It is also curious that the pnp-1 mutants resemble the pals-22 mutant to a substantial extent. I wonder if the authors could address this with some experiments.

Reviewer #3: In this manuscript, Tecle and co-authors describe a new role for purine metabolism in regulating defense responses against intestinal pathogens in C. elegans. Through a forward genetic screen the authors have identified the enzyme purine nucleoside phosphorylase, pnp-1, as a negative regulator of the intracellular pathogen response (IPR). Consequently, pnp-1 loss of function mutants were found to be more resistant to Orsay virus and N. parisii infection just like the previously described mutants of pals-22, which is another negative regulator of IPR. However, pnp-1 mutants did not exhibit some phenotypes reported for pals-22 mutants, like reduced lifespan and increased thermotolerance. Also, pnp-1 mutants showed mild resistance towards bacterial infection by PA which was contrasting to increased susceptibility reported for pals-22 mutants previously. While the exact mechanistic link between purine metabolism and epithelial cell defence is not yet understood, the study is well-written and contains novel and convincing results that are likely to be generalisable to other systems. As such, I believe the manuscript will be of interest to readers of PloS Pathogens. Here are a few points that can help the authors to further strengthen their manuscript.

**Part II – Major Issues: Key Experiments Required for Acceptance**

Reviewer #1: (No Response)

Reviewer #2: - The paper describes the role of pnp-1 well, but some metabolite supplementation experiments might be able to better pinpoint what drives the gene expression and consequent organismal phenotypes. Specifically, the authors should test whether supplementation with hypoxanthine rescues some of the key defects of the pnp-1 mutants (or just one). This would reveal whether the defects arise due to lack of hypoxanthine (in which case they should be rescued) or due to build-up of inosine (in which case they might not be). One might hypothesize that the former would the the case as hypoxanthine availability might be limiting.

- Along those lines, what are the effects of hypoxanthine and inosine supplementation on gene expression. While RNA-seq is certainly outside of the scope of the manuscript, qRT-PCR of a few key genes (pals, etc) or even just a pals-5p::GFP reporter study in hypoxanthine-supplemented mutants and inosine-supplemented WT worms would reveal whether low hypoxanthine or high inosine drive the gene expression changes.

- Further to this, the authors mention, but don’t discuss in detail, that no other metabolites in purine metabolism are changed. This is intriguing: how do the very substantial changes in inosine and hypoxanthine nevertheless not affect any of the other metabolites in this cycle? Does this reflect reduced rate in this metabolic pathway, i.e., steady state levels are the same but flux through the pathway is reduced (Fig S2 beautifully shows that there really are no apparent major detectable changes elsewhere)? I would be interested in the authors opinions on this.

- I’m also intrigued by the similarity between the pnp-1 mutants and the pals-22 mutant. How is this possible - do they regulate each other (unlikely as shown by metabolomics)? If not, what else can explain how a pnp-1 mutant results in significant similarity to the mutation in a pals gene? Again, I’d be interested in the authors opinion.

- Fig 5 contains lots of information that could be mined some more. For example, the genes that are regulated in both pnp-1 mutants but not in the pals-22 mutant should be the ones that drive the extracellular pathogen response phenotype - can this be ascertained in some way? I.e. the authors also write that the pnp-1 genes are also significantly similar to genes regulated sta-1, skn-1, hsf-1, pmk-1 and sek-1, which in turn were different from IPR genes; this similarity should derive from that same gene set of 84 genes in Fig 5B - does this contain genes with e.g. predicted binding sites for any of these TFs and/or genes already known from other (screen) papers to confer resistance to infection with gram negative bacteria? etc.

- Methods: please, please deposit the RNA-seq results in NCBI GEO. This is a very interesting dataset and other investigators would greatly benefit from readily accessible raw data.

Reviewer #3: The authors have shown that loss of pnp-1 leads to accumulation of inosine and activation of IPR in independent experiments. Can they show that direct supplementation of inosine is sufficient to trigger activation of IPR or make worms less susceptible to Orsay virus or N. parisii infection? The same also applies to supplementation of hypoxanthine, which is proposed to act as a repressor of pathogen response gene expression. Perhaps the authors have already tried these experiments so it may be good to discuss.

To strengthen their model that PALS-22 and PNP-1 act in parallel, can the authors show that the described IPR changes in pnp-1(-) mutants are not dependent on PALS-25?

**Part III – Minor Issues: Editorial and Data Presentation Modifications**

Reviewer #1: - Figure 6: The authors suggest that the host may sense alterations in purine metabolism caused by pathogen infection as part of ‘surveillance immunity’. Here, perturbations in core cellular processes can lead to activation of pathogen defenses, like in pnp-1 mutants that exhibit alterations in purine metabolism and activation of the IPR and extracellular pathogen response genes. According to this model pnp-1 affects gene expression only indirectly. This does not become clear in the concluding model shown in Figure 6. Also, shouldn’t the arrows from the pathogen infection point to purine metabolism rather than directly to the gene cassettes? Spelling: Microsporisia infection > Microsporidia infection

- Do pnpn-1 mutants exhibit increased survival upon infection with P. aeruginosa?

- All statistical tests used are parametric tests. Did the authors check if their data are really all normally distributed?

Reviewer #2: - In general it would be good if the authors would replace bar graphs with dot graphs or box and whisker graphs that better reflect the measurements. Fig 2 especially features graphs with large variation and it would be great to see individual data points there and elsewhere, as shown in Fig 2C and E.

- Fig 1E, the authors study several genes by qPCR. I noted the accession numbers F26F2.1, .3, and .4 - are these in an operon? If so, it’s natural that they display similar expression patterns. Please clarify.

- L167 "N. parsii”

- Fig 3B may have an error with stats comparison, i.e. is the difference between pnp-1 mutant (2nd group) and non-rescued pnp-1 mutant really significant at **? Likely meant to be comparison between the latter two groups. Moreover, Fig 3B-C, the comparison between group 1 (WT) and group 3 (mutant rescued with WT) stats should also be shown.

- L258 & Fig 5 the authors write "25 pals genes upregulated by N. parisii infection and Orsay virus, 22 are significantly up-regulated” but the figure shows only 13 genes.

Reviewer #3: The authors show that pals-22 mutants have higher resistance to viral/microsporidial infection than pnp-1 mutants. Is there any evidence that IPR induction in pnp-1 mutants is overall milder compared to that observed in pals-22 mutants based on qPCR (Fig. 1E) or RNAseq (Fig. 5)?

It may be useful to show the overlap with IPR in the Venn diagrams in Figure 5A,B

PLOS authors have the option to publish the peer review history of their article (what does this mean?). If published, this will include your full peer review and any attached files.

Reviewer #1: **Yes: **Katja Dierking

Reviewer #2: No

Reviewer #3: No

Figure Files:

Data Requirements:

Reproducibility:

References:

---

## [Editor Report · Decision Letter 1]

6 Apr 2021

Dear Dr. Troemel,

We are pleased to inform you that your manuscript 'The purine nucleoside phosphorylase pnp-1 regulates epithelial cell resistance to infection  in C. elegans' has been provisionally accepted for publication in PLOS Pathogens.

Best regards,

James J Collins III

Section Editor

PLOS Pathogens

James Collins III

Section Editor

PLOS Pathogens

Kasturi Haldar

Editor-in-Chief

PLOS Pathogens

orcid.org/0000-0001-5065-158X

Michael Malim

Editor-in-Chief

PLOS Pathogens

orcid.org/0000-0002-7699-2064
---

## [Editor Report · Acceptance letter]

15 Apr 2021

Dear Dr. Troemel,

We are delighted to inform you that your manuscript, "The purine nucleoside phosphorylase pnp-1 regulates epithelial cell resistance to infection  in C. elegans," has been formally accepted for publication in PLOS Pathogens.

Best regards,

Kasturi Haldar

Editor-in-Chief

PLOS Pathogens

orcid.org/0000-0001-5065-158X

Michael Malim

Editor-in-Chief

PLOS Pathogens

orcid.org/0000-0002-7699-2064